# Transcriptomic and Translatomic Analyses Reveal Insights into the Signaling Pathways of the Innate Immune Response in the Spleens of SPF Chickens Infected with Avian Reovirus

**DOI:** 10.3390/v15122346

**Published:** 2023-11-29

**Authors:** Sheng Wang, Tengda Huang, Zhixun Xie, Lijun Wan, Hongyu Ren, Tian Wu, Liji Xie, Sisi Luo, Meng Li, Zhiqin Xie, Qing Fan, Jiaoling Huang, Tingting Zeng, Yanfang Zhang, Minxiu Zhang, You Wei

**Affiliations:** 1Guangxi Key Laboratory of Veterinary Biotechnology, Guangxi Veterinary Research Institute, Nanning 530000, China; wangsheng1021@126.com (S.W.); wanlijun0529@163.com (L.W.); renhongyu328@126.com (H.R.); xie3120371@163.com (L.X.); 2004-luosisi@163.com (S.L.); mengli4836@163.com (M.L.); xzqman2002@sina.com (Z.X.); fanqing1224@126.com (Q.F.); huangjiaoling728@126.com (J.H.); tingtingzeng1986@163.com (T.Z.); zhangyanfang409@126.com (Y.Z.); zhminxiu2010@163.com (M.Z.); weiyou0909@163.com (Y.W.); 2Key Laboratory of China (Guangxi)-ASEAN Cross-Border Animal Disease Prevention and Control, Ministry of Agriculture and Rural Affairs of China, Nanning 530000, China; 3Division of Liver Surgery, Department of General Surgery, Laboratory of Liver Surgery, and State Key Laboratory of Biotherapy, West China Hospital, Sichuan University, Chengdu 610041, China; 2022324065113@stu.scu.edu.cn; 4NHC Key Laboratory of Transplant Engineering and Immunology, Regenerative Medicine Research Center, Frontiers Science Center for Disease-Related Molecular Network, West China Hospital of Sichuan University, Chengdu 610041, China; wutian980518@126.com

**Keywords:** translatomics, Ribo-seq, avian reovirus, spleen, innate immunity, IL4I1

## Abstract

Avian reovirus (ARV) infection is prevalent in farmed poultry and causes viral arthritis and severe immunosuppression. The spleen plays a very important part in protecting hosts against infectious pathogens. In this research, transcriptome and translatome sequencing technology were combined to investigate the mechanisms of transcriptional and translational regulation in the spleen after ARV infection. On a genome-wide scale, ARV infection can significantly reduce the translation efficiency (TE) of splenic genes. Differentially expressed translational efficiency genes (DTEGs) were identified, including 15 upregulated DTEGs and 396 downregulated DTEGs. These DTEGs were mainly enriched in immune regulation signaling pathways, which indicates that ARV infection reduces the innate immune response in the spleen. In addition, combined analyses revealed that the innate immune response involves the effects of transcriptional and translational regulation. Moreover, we discovered the key gene IL4I1, the most significantly upregulated gene at both the transcriptional and translational levels. Further studies in DF1 cells showed that overexpression of IL4I1 could inhibit the replication of ARV, while inhibiting the expression of endogenous IL4I1 with siRNA promoted the replication of ARV. Overexpression of IL4I1 significantly downregulated the mRNA expression of IFN-β, LGP2, TBK1 and NF-κB; however, the expression of these genes was significantly upregulated after inhibition of IL4I1, suggesting that IL4I1 may be a negative feedback effect of innate immune signaling pathways. In addition, there may be an interaction between IL4I1 and ARV σA protein, and we speculate that the IL4I1 protein plays a regulatory role by interacting with the σA protein. This study not only provides a new perspective on the regulatory mechanisms of the innate immune response after ARV infection but also enriches the knowledge of the host defense mechanisms against ARV invasion and the outcome of ARV evasion of the host’s innate immune response.

## 1. Introduction

Avian reovirus (ARV) is part of the genus Orthoreovirus in the Spinareoviridae family, and the main symptoms it causes include viral arthritis, chronic respiratory disease, growth retardation and malabsorption syndrome. In addition, ARV infection can cause severe immunosuppression and predispose patients to other complications or secondary infections. ARV infection is widespread in the global poultry industry; vaccination is mainly used to prevent it in the fowl industry, but it is still not well prevented or controlled. ARV infection causes major economic losses for farmers [1,2,3].

At present, good progress has been made in the research and development of ARV vaccines and the establishment of detection methods, but research on the pathogenic mechanism of ARV and the antiviral response of host innate immunity is still in the development stage. Furthermore, clarifying the interaction mechanism of ARV is necessary for its prevention, control and treatment.

Innate immunity is the body’s first line of defense against ARV invasion. The body activates the production of inflammatory factors and interferons by recognizing pathogenic pattern-related molecules through pattern recognition receptors and activates acquired immunity to trigger a comprehensive immune response [4,5]. Many studies have been conducted on the regulation of innate immune-related pattern recognition receptors and their effector factors during ARV infection. In the early period of ARV infection, the PI3K/Akt/NF-κB and STAT3 signaling pathways can be activated to induce an inflammatory response [6,7]. Lostalé-Seijo et al. found that ARV infection of chicken embryonic fibroblasts induced the expression of interferon and interferon-stimulated genes Mx and double-stranded RNA-dependent protein kinases (PKRs) to exert antiviral effects [8].

In the early stage of ARV infection, the virus can induce the activation of MDA5 signaling pathway-related molecules in chicken peripheral blood lymphocytes, thereby inducing the production of inflammatory factors and interferons [9]. In a previous study, we detected the transcriptional expression level changes in interferon and interferon-stimulated genes (Mx, IFITM3, IFI6 and IFIT5) in several different tissues and organs of ARV-infected specific pathogen-free (SPF) chickens using real-time PCR, and the results suggested that ARV infection can cause significant changes in these effector factors, indicating that the process of ARV infection is closely linked to the recognition of host innate immune-related model receptors and the production of effector factors [10,11]. However, the limited number of genes that were tested made it impossible to fully characterize the gene regulation of innate immunity during ARV infection.

Next-generation sequencing technology has become an effective tool for studying the interaction between viruses and hosts. However, due to the poor correlations between mRNA abundance and protein abundance, traditional mRNA sequencing (RNA-seq) cannot accurately depict the whole realm of gene expression, particularly in regards to reflecting the actual expression levels of proteins [12]. Ribosome profiling, also known as ribosomal footprint sequencing (Ribo-seq), is a new high-throughput sequencing technology developed in recent years that sequences ribosome-protected mRNA fragments (RPFs) [13]. Ribo-seq can accurately measure translational activity and abundance genome-wide [14], making it possible to investigate the ribosome density profile of the translatome [15,16]. Association analysis combined with RNA-seq and Ribo-seq methods can be used to study post-transcriptional regulation and translational regulation mechanisms. Ribo-seq has been used to explore the mechanisms of translation regulation in different species, such as humans [17], mice [18], zebrafish [19], Drosophila [20], rice [21], Arabidopsis [22] and maize [23]. However, the translational regulation mechanisms in chickens remain poorly studied.

The spleen is the largest and most important peripheral immune organ in chickens, and it plays a major role in maintaining the balance of immune function and evading the invasion of pathogenic microorganisms. Previous studies have found that ARV infection can cause harm to the spleen, which leads to immunosuppression [24,25]. The results of our previous study showed that the viral load in the spleen after ARV infection was noticeably above those in the thymus and bursa of Fabricius, suggesting that the spleen is the main immune organ attacked by ARV. In addition, the mRNA expression of various interferon-stimulated genes in the spleen after ARV infection was rapidly upregulated in the early stage of infection, indicating that ARV infection can induce a strong innate immune response in the spleen. Analysis of the pathological changes in the spleen after ARV infection showed that there were no obvious lesions on days 1 to 2, while generalized necrotic degeneration of the lymphocytes and homogeneous red staining of the splenic body were observed on day 3. This pathological injury continued until day 7 and was gradually relieved. Interestingly, the ARV viral load in the spleen remained high for 1 to 3 days after infection and then decreased sharply on day 4 [11]. Therefore, it is speculated that the early stage of ARV infection, especially day 3, is a critical period for ARV invasion of the spleen. Transcriptional and translational regulation play major roles in host innate immunity against viral infection. In this research, SPF chickens artificially infected with the ARV S1133 strain were used as subjects. Their splenic tissues were dissected on the third day after infection, and RNA-seq and Ribo-seq analyses were performed to study chicken spleen gene regulation after ARV infection at the transcriptional and translational levels. Finally, functional genes that play an important part in the innate immune response were identified, and their functions were further analyzed. Our results provide a comprehensive understanding of the immune evasion of ARVs in the spleen and of the host immune defense against ARVs.

## 2. Results

### 2.1. Overview of High-Throughput Sequencing Data between the Spleens of Control Group and ARV SPF Chickens

To explore genome-wide innate immune response regulation from a translational perspective, we compared the ribosomal maps of the spleens of control group (CON) and ARV SPF chickens using ribosomal footprint sequencing and mRNA sequencing. The Ribo-seq and RNA-seq libraries of the CON and ARV groups were prepared and sequenced on HiSeq-2000 platforms, resulting in 13.1–14.8 million and 16–16.2 million ribosome profiling clean reads for the CON and ARV groups, as well as 17.3–17.8 million and 14.5–16.5 million RNA-seq clean reads for the CON and ARV groups, respectively (Appendix A). RNA-seq identified and quantified 17,721 genes and 17,693 genes in the CON and ARV groups, respectively (Appendix A). Ribo-seq identified and quantified 15,760 genes and 14,776 genes in the CON and ARV groups, respectively (Appendix A), and their expression abundance levels all had a similar normal distribution. Appendix A display the gene expression abundance information of the transcriptome and translatome. The Pearson correlation analysis of RNA-seq and Ribo-seq exhibited similarities and differences between the CON and ARV groups (R^2^ > 0.92, Appendix A). These results indicate that subsequent analyses are performed on the basis of reliable data.

### 2.2. Global Translatome Characteristics

To investigate whether the characteristics of the ribosome-protected fragments change with innate immune response regulation in the spleen, the basic ribosome profiles of RPFs were compared between the CON and ARV groups. The length distribution of RPF peaks at 28 nt for both the CON and ARV groups (Figure 1A). Figure 1B shows that the distribution patterns of RPFs for the CON and ARV groups were similar, with the vast majority of ribosome footprints located at the CDS of both CON and ARV mRNAs. These results are similar to those in most eukaryotes, suggesting that the translation process is highly conserved [26]. However, compared to the CON group, the RPFs of the ARV group in the CDSs and 5′ UTRs decreased to 3.87% and 3.31%, respectively. The ARV RPFs in 3′ UTRs increased from 8.02% to 15.19%. These results suggest that the apparent activation of RPFs in 3′ UTRs may be caused by ARV infection in chickens. In addition, three-nucleotide periodicity was clearly observed around the start and stop codon regions of RPFs at different read lengths of 28 nt. A vast number of RPFs were enriched in the region approximately from position −12 nt to the annotated start codon (Figure 1C and Appendix A), indicating that the initiation stage is the principal rate-limiting stage of translation.

### 2.3. Translational Efficiency Significantly Decreased after ARV Infection

TE is an important index of translation that reflects the efficiency of mRNA utilization, and the formula is as follows: TE = (RPKM from translatome)/(FPKM from transcriptome) [14,27]. The average TE of genes across the whole genome decreased significantly after ARV infection (log2 mean TE of CON = −0.09297, log2 mean TE of ARV = −0.6425; Figure 2A). In addition, the ratio of genes with higher TE (log2TE > 1) in the ARV group was lower than that in the CON group (Figure 2B). Moreover, there were 15 upregulated, differentially expressed TE genes and 396 downregulated, differentially expressed TE genes after ARV infection compared to the CON group (Figure 2C). In regard to the functions of the differentially expressed TE genes, GO and KEGG analyses were conducted. The GO analysis displayed the top 20 terms, which mainly included cell activation, regulation of response to stimulus, regulation of immune system process, etc. (Figure 2D). The KEGG pathway enrichment analysis showed the top 20 pathways for which gene expression was enriched. Among the top 20 pathways, 7 pathways belonged to the “immune system” category, including the chemokine signaling pathway, Fc epsilon RI signaling pathway, T-cell receptor signaling pathway, intestinal immune network for IgA production, complement and coagulation cascades, B-cell receptor signaling pathway and hematopoietic cell lineage. The rheumatoid arthritis pathway was also among the top 20 pathways and belonged to the “immune diseases” category (Figure 2E). The above results indicate that after infection with ARV, gene translation efficiency is reduced at the overall level, and the translation efficiency of immune-related genes is significantly affected.

### 2.4. Regulation Patterns of the Transcriptome and Translatome

Based on both ribosome profiling and RNA sequencing data, the transcriptional and translational expression differences between the CON and ARV groups were examined. The CON and ARV groups had high correlations for transcriptome and translatome (R^2^ = 0.8364 in RNA-seq, R^2^ = 0.8531 in Ribo-seq; Appendix A), which illustrates that transcriptome analysis and translatome analysis are reliable. The differentially expressed genes in both the RNA-seq and Ribo-seq data sets were filtered based on the criteria of |log2 fold change| > 1 and FDR < 0.01. There were 225 transcriptionally upregulated and 439 downregulated DEGs in the ARV group compared to the CON group, corresponding to 851 upregulated and 1128 downregulated DEGs at the translational level (Appendix A). The quantities of downregulated genes were much greater than those of the upregulated genes at two levels, suggesting a global decline in gene expression in the ARV group.

To explore the relationships and differences in the regulation of gene expression in the spleen at the transcriptional level and the translation level after ARV infection, we conducted a combined analysis of the transcriptome and the translatome. Figure 3A displays the scatter plot of the fold changes in transcriptional and translational expression. The scatter plot was classified into nine categories based on the criteria of |log2 fold change in RPKM| > 1 and FDR < 0.01. The gene information of the nine categories can be found in Appendix A. Our results revealed that 81.96% of genes were categorized in the unchanged class (quadrant E), and 14.7% of genes were in the discordant classes (quadrants A, B, D, F, H, I). Notably, 1.31% (154) and 2.02% (238) of genes were located in quadrants C and G, respectively, which meant that the expression of genes changed congruously at the transcriptional and translational levels (upregulation for quadrant C; downregulation for quadrant G). Furthermore, to explore the synergistic functions of transcription and translation, GO analysis of the biological processes enriched for the congruous DEGs (quadrants C and G) was conducted. The results showed that the congruous DEGs were significantly enriched for terms related to innate immunity such as cell surface receptor signaling pathway, immune response, innate immune response, immune system process and regulation of immune system process (Figure 3B). These results suggest that the innate immune response of the body to ARV infection involves co-regulation of transcription and translation.

### 2.5. Screening Functional Genes after ARV Infection

The above association analysis showed that genes in quadrants C and G were mainly enriched for biological processes related to immune regulation, indicating that these DEGs may be involved in the innate immune response to ARV infection. We identified 392 DEGs in quadrants C and G. We compared the top 30 DEGs at the transcriptional and translational levels (Figure 4A,B). Based on the significance of the DEGs, the most significant DEG at the transcriptional and translational levels was determined to be IL4I1 (FDR in transcriptome = 1.85 × 10^−27^, FDR in translatome = 6.2 × 10^−57^). According to the RNA-seq and Ribo-seq data, the expression level of IL4I1 significantly increased after ARV infection (Figure 4C). The RT-qPCR results were consistent with the RNA-seq and Ribo-seq results (Figure 4D).

### 2.6. IL4I1 Expression Reduced ARV Replication

To verify the effect of IL4I1 overexpression on ARV replication, we transfected the pEF1α-Myc-IL4I1 recombinant plasmid into DF1 cells, then verified IL4I1 overexpression by real-time PCR and Western blotting 24 h later. Real-time PCR showed that IL4I1 gene expression in DF1 cells was significantly upregulated after transfection with the pEF1α-Myc-IL4I1 (Figure 5A). Detection at approximately 60 kDa using Myc-tagged antibodies showed that IL4I1 was correctly expressed in DF1 cells transfected with the pEF1α-Myc-IL4I1, while IL4I1 protein expression was not detected in cells transfected with empty vectors (Figure 5B). Cells were infected with ARV after 24 h of transfection, and cell samples and supernatants were collected after another 24 h. Real-time PCR detection showed that the expression of the ARV σC gene at the mRNA level was significantly reduced (Figure 5C), and the ARV virus titer in the cell supernatant was also significantly reduced (Figure 5D). These results suggest that the overexpression of IL4I1 in DF1 cells inhibits the replication of ARV.

Therefore, we speculated that IL4I1 can cause negative feedback in the replication of ARV and that inhibiting the expression of IL4I1 can promote the replication of ARV. We designed and synthesized three siRNAs against the IL4I1 gene to inhibit the expression of IL4I1, of which siRNA1357 was the most ideal (Figure 5E). siRNA1357 was transfected into DF1 cells, and the cells were infected with ARV virus after 24 h of transfection. Then, cell samples and supernatants were collected at 24 h postinfection to detect ARV replication at the gene expression and viral titer levels, and the results were consistent with expectations (Figure 5F,G). The above results show that inhibiting the expression of IL4I1 can promote the replication of ARV.

### 2.7. The Effect of IL4I1 Expression on the Innate Immune Response during ARV Infection

To investigate how IL4I1 regulates the innate immune response induced by ARV infection, we overexpressed or inhibited IL4I1 and detected the effect of IL4I1 expression on the expression of innate immune signaling pathway-correlated factors during ARV infection by real-time PCR. The results showed that the mRNA expression of MDA5, TRAF3 and TRAF6 was significantly upregulated after overexpression or inhibition of IL4I1. The mRNA expression of MAVS was upregulated after overexpression of IL4I1 and downregulated after inhibition of IL4I1. The mRNA expression of IKKε did not differ significantly after overexpression of IL4I1 and was significantly upregulated after inhibition of IL4I1. The mRNA expression of IRF7 was significantly upregulated after overexpression of IL4I1, but there was no significant difference after IL4I1 inhibition. There was no significant difference in the mRNA expression of IFN-α after overexpression or inhibition of IL4I1. The mRNA expression of IFN-β, LGP2, TBK1 and NF-κB was significantly downregulated after overexpression of IL4I1 and upregulated after inhibition of IL4I1 (Figure 6A,B). These results suggest that IL4I1 may be a negative feedback regulator of innate immune signaling pathways and that IL4I1 expression may reduce IFN-β production by inhibiting the expression of LGP2, TBK1 and NF-κB.

### 2.8. Interaction between IL4I1 and ARV σA/σC Proteins

The σA and σC proteins are important structural proteins of ARV and play a significant part in the interaction between ARV and the host. Therefore, we studied the relationship between IL4I1 and ARV σA/σC. We transfected the eukaryotic expression plasmids pEF1α-HA-σA and pEF1α-HA-σC into DF1 cells and overexpressed ARV σA and σC proteins in DF1 cells. Real-time PCR detection showed that the expression of IL4I1 was significantly upregulated after the overexpression of σA and σC proteins in DF1 cells. The overexpression of σA in particular upregulated IL4I1 by a relatively high fold change (Figure 7A).

Subsequently, we used Co-IP to determine whether IL4I1 interacted with ARV σA and σC proteins in vitro. The Co-IP results of IL4I1 protein and ARV σA protein showed that when Co-IP immobilization used the anti-Myc monoclonal antibody, the σA-HA protein could be identified by Western blot. However, when Co-IP immobilization used the anti-HA monoclonal antibody, the IL4I1-Myc protein was not identified by Western blot (Figure 7B).

The Co-IP results of IL4I1 protein and ARV σC protein showed that when Co-IP immobilization used the anti-Myc monoclonal antibody, no σC-HA protein was identified by Western blot. IL4I1-Myc protein was also not identified by Western blot when Co-IP immobilization used the anti-HA monoclonal antibody (Figure 7C). Therefore, there may be an interaction between the IL4I1 and ARV σA proteins.

## 3. Discussion

Gene expression is closely related to the occurrence and development of various physiological and pathological activities and diseases. High-throughput sequencing technology enables the sequencing and identification of millions of nucleotide molecules simultaneously. To date, it is widely used in the screening of important functional genes and research on animal diseases [28]. Translation regulation is a key element in the regulation of gene expression. Omics studies have shown that translation regulation accounts for more than half of all regulation overseeing gene expression, and the translational differences better reflect the expression changes in the proteome than those of the transcriptome [29,30].

In this study, Ribo-seq was conducted to reveal the gene expression profile of the spleen after ARV infection. We analyzed the characteristics of RPFs in the spleen; after ARV infection, the abundance of splenic RPFs in the CDS region and the 5’ UTR was lower than that in the control group, which also indicated that ARV infection inhibited the process of protein synthesis in the spleen. After ARV infection, the abundance of splenic RPFs in the 3’ UTR was higher than that in the CON group. In the eukaryotic translational process, the 5’ UTR and the 3’ UTR play important roles in post-transcriptional regulation. The 5’ UTR mediates post-transcriptional regulation through the main elements present in this region, such as uORFs, secondary structures and RPF-binding motifs, and the 3’ UTR contains a large number of regulatory elements, such as microRNA binding sites and protein binding sites [31,32,33]. After ARV infection, the abundance of splenic RPFs increased significantly in the 3’ UTR, suggesting that there may be potential translational regulation of ARV infection in the 3’ UTR, which may be related to the innate immune regulation process of the host after ARV infection. These complex regulatory mechanisms still need to be further studied.

Studies have confirmed that genes with higher translation efficiency perform more important biological functions, and in addition, the specific array of genes with high translation efficiency or upregulated translation efficiency reflect the function and phenotype of a particular cell [14]. Our sequencing results showed that ARV infection significantly reduced the overall translation efficiency of splenic genes. Further analysis showed that, compared with the control group, there were 15 significantly upregulated DTEGs and 369 significantly downregulated DTEGs in the ARV infection group. These DTEGs were mainly enriched in signaling pathways related to immune regulation, which led to speculation that ARV infection would reduce the immune response ability of the spleen. Recent studies have confirmed that ARV infection causes immunosuppression [34]. However, the specific mechanism of ARV-induced immunosuppression still needs to be studied in greater depth. After ARV infection, the translation efficiency of immune regulation-related signaling pathways in the spleen is reduced, which we speculate is one of the important reasons for immunosuppression caused by ARV infection.

The transcriptome and translatome association data analysis showed that 392 genes were expressed in common with significant differences at the transcription and translation levels. GO enrichment analysis showed that the 20 most significant GO terms were all enriched in signaling pathways related to immune regulation. This suggests that these DEGs play important roles in the innate immune response to ARV infection. We further analyzed the significance of these 392 congruous DEGs and screened the 30 genes with the highest significance. The most significantly differentially expressed gene was IL4I1 in both the transcriptome and translatome. The expression of IL4I1 was highly induced after ARV infection, suggesting that IL4I1 plays an important part in the response to ARV infection. Interleukin-4-induced-1 (IL4I1) is a less-studied amino acid catabolic enzyme that belongs to the L-amino acid oxidase family. IL4I1 plays an important role in the body’s defense against infection, regulation of immune homeostasis and injury response. In recent years, it has been found that IL4I1 is closely related to the regulatory process of human immune metabolism, and it is an important immunosuppressive factor and a key metabolic immune checkpoint [35]. Tumor cells produce large amounts of the IL4I1 metabolic enzyme, which promotes the spread of tumor cells and suppresses the immune system. IL4I1 breaks down tryptophan to form indole metabolites and kynurequinolinic acid, which are agonists of the aryl hydrocarbon receptor (AHR). Indole metabolites and kynurequinolinic acid bind to and activate AHR receptors, thereby mediating the toxic effects of dioxins, which reduces the utilization of essential or semi-essential amino acids. At the same time, toxic metabolites are produced, which cause damage to antitumor T lymphocytes and promote the growth of tumor cells [36].

There are few reports on the role and mechanism of IL4I1 in avian virus infection. Studies by high-throughput sequencing technology have found that a large amount of IL4I1 expression can be induced after a variety of viral infections. Hu et al. found that IL4I1 expression was significantly upregulated in chicken embryo fibroblasts infected by the J subpopulation of avian leukemia virus (ALV-J) by transcriptome sequencing [37]. Feng et al. used transcriptome sequencing analysis to find that the expression of IL4I1 in chicken primary mononuclear macrophages infected with ALV-J was significantly upregulated 3 h and 36 h after infection, and overexpression of IL4I1 at the gene level could facilitate the replication of ALV-J in chicken macrophages [38]. Dong et al. found that the IL4I1 gene was upregulated in chicken spleen tissues infected with Marek virus (MDV) by transcriptome sequencing technology [39]. Conversely, our study showed that IL4I1 inhibits the replication of the ARV virus. This suggests that there may be differences in the roles of IL4I1 in the replication of different viruses. We found that IL4I1 was able to prevent the replication of ARV in DF1 cells by overexpression or inhibition assays. Our previous studies showed that the ARV viral load in the spleen was high on days 1 to 3 after ARV infection and then decreased sharply on day 4 [11]. Therefore, we speculate that the rapid upregulation of IL4I1 expression in the spleen after ARV infection is beneficial to inhibiting the proliferation of ARV in the spleen. However, IL4I1 was able to promote the replication of ALV-J in chicken macrophages, suggesting that there may be differences in the roles of IL4I1 in the replication of different avian viruses. We also detected a regulatory role for IL4I1 on innate immune signaling pathway-correlated factors during ARV infection by real-time PCR. The mRNA expression of IFN-β, LGP2, TBK1 and NF-κB was significantly downregulated after overexpression of IL4I1, and the mRNA expression of IFN-β, LGP2, TBK1 and NF-κB was significantly upregulated after inhibiting IL4I1. This suggests that IL4I1 may be a negative feedback effect of innate immune signaling pathways during ARV infection, and IL4I1 may reduce the production of IFN-β by inhibiting the expression of LGP2, TBK1 and NF-κB. The current study has shown that IL4I1 is an important immunosuppressive molecule that plays a key role in the immune evasion of tumors [36]. Previous studies have also found that IL4I1 can inhibit the production of IFN-γ and inflammatory cytokines, limit local Th1 inflammation and inhibit the inflammatory response [40]. ARV is an important avian immunosuppressive disease, and our sequencing analysis found that ARV infection can significantly reduce the translational efficiency of immunomodulatory-related genes in chicken spleen. ARV, ALV-J and MDV infection can cause avian immunosuppressive diseases, and IL4I1 expression was upregulated after each of these three viral infections. Whether the regulation of IL4I1 expression is related to the immunosuppressive processes caused by avian immunosuppressive viruses is an interesting mechanism that deserves more in-depth study.

Virus-encoded proteins play a crucial part in virus interactions with its host. The σA protein of ARV plays a vital role in the pathogenesis of ARV infection. A study found that the ARV σA protein binds irreversibly to viral dsRNA, thereby inhibiting the dsRNA-dependent protein kinases activation and ultimately interfering with the antiviral effects of interferon [41,42]. In addition, the ARV σA protein can also activate the PI3K/Akt signal transduction pathways in cells, increase the expression of phosphorylated Akt (p-Akt) in cells and, thus, inhibit the apoptosis of infected cells to facilitate ARV infection and replication [43]. It has also been showed that the ARV σA protein affects the replication of ARV in DF1 cells by interacting with the NME2 protein of the host [44]. The ARV σC protein is related to the adsorption and proliferation of virions [45]. The ARV σC protein is able to induce apoptosis by interacting with the host protein EFF1A1 [46]. In this study, we found that overexpression of ARV σA and σC proteins in DF1 cells can cause significant upregulation of IL4I1 expression at the transcriptional level. The interaction between IL4I1 and ARV σC proteins was not found by co-immunoprecipitation experiments, while the co-immunoprecipitation analysis of IL4I1 protein and ARV σA proteins uncovered an interesting phenomenon. When we used the anti-Myc antibody to fix IL4I1 protein in the Co-IP experiment, σA protein interacted with IL4I1 protein, but when we fixed σA protein using the anti-HA antibody in the Co-IP experiment, the interaction between σA protein and IL4I1 protein could not be detected. To ensure the rigor of the experimental data, we performed multiple replicates using antibodies and Co-IP kits of different brands, all with the same results. We reviewed the literature and found that the human IL4I1 protein is a glycosylated secreted protein [47]. The structure and function of avian IL4I1 protein have not been reported, and our online software analysis shows that avian IL4I1 protein is also a secreted protein. We hypothesize that the IL4I1 protein failed to be detected by Western blot analysis when the σA protein was immobilized in the Co-IP experiment, which is related to the fact that IL4I1 is secreted extracellularly after synthesis. Therefore, we speculate that there may be an interaction between the IL4I1 protein and the ARV σA protein, and that the IL4I1 protein may actively bind to the σA protein. We speculate that the IL4I1 protein may play a regulatory role by interacting with the ARV σA protein. We further speculate that after ARV infection, IL4I1 is modulated and then transcribed and expressed in large quantities, and the IL4I1 protein competitively binds to the ARV σA protein, thereby affecting the function of the ARV σA protein and inhibiting the replication of ARV. However, due to a lack of avian-derived IL4I1 protein-specific antibodies, it is difficult to further verify the interaction between the IL4I1 and ARV σA proteins. In future studies, we will prepare monoclonal antibodies against avian IL4I1 protein and conduct in-depth research on the role and regulatory mechanism of IL4I1 in the interaction network of ARV or σA protein.

## 4. Materials and Methods

### 4.1. Ethics Statement

This study was approved by the Animal Ethics Committee of Guangxi Veterinary Research Institute. Animal experiments and sample collection were conducted in accordance with the guidance of protocol #2019C0406 issued by the Animal Ethics Committee of Guangxi Veterinary Research Institute.

### 4.2. Viral Inoculations and Animal Experiments

The ARV S1133 strain used in the study was purchased from the China Institute of Veterinary Drug Control. “White Leghorn” SPF chicken eggs were purchased from Beijing Boehringer Ingelheim Vital Biotechnology Co., Ltd. (Beijing, China). Incubation was performed using a fully automated incubator, after which chicks were raised in SPF chicken isolators. A total of twenty 7-day-old SPF chickens were randomized into two groups and raised aseptically in an SPF chicken isolator. Group A was the experimental group (ARV), and each chicken was inoculated with 0.1 mL 10^4^ TCID_50_/0.1 mL ARV S1133 virus by foot pad injection. Group B was the control group (CON), which was inoculated with the same amount of PBS via foot pad injection. Samples were collected on day 3 after infection. The chickens were taken from the ARV infection group and the control group for dissection and collection of spleens, and then the collected samples were snap-frozen in liquid nitrogen and stored in a −80 °C freezer for subsequent analysis. The standard for selecting sequencing samples in this study was to use the two samples whose viral load of spleen was closest to the mean in the group as 2 biological replicates.

### 4.3. RNA Extraction and Transcriptome Sequencing

Total RNA of the spleen in each group was extracted by using TRIzol^®^ RNA extraction reagent (Invitrogen, Carlsbad, CA, USA) according to the manufacturer’s instructions. The integrity of RNA was examined by agarose gel electrophoresis, and the concentration of RNA was measured by NanoDrop 2000 spectrophotometers (Thermo Fisher Scientific, Boston, MA, USA). The mRNA was purified by oligo (dT) magnetic beads and fragmented into short fragments using fragmentation buffer. First-strand cDNA was synthesized with SuperScript II Reverse Transcriptase (Invitrogen) using random primers, and second-strand cDNA was synthesized using the synthesized first strand of cDNA as a template. The obtained double-stranded cDNA was purified by a VAHTS^®^ mRNA-seq V3 Library Prep Kit for Illumina (Vazyme, Nanjing, China), end repaired, poly(A) added and then ligated to Illumina sequencing adapters. Sequencing was performed on the Illumina HiSeq-2000 platform for 50 cycles. High-quality reads passed through the Illumina quality filter were retained in fastq.gz format for sequence analysis.

### 4.4. Preparation of Ribosome-Protected Fragments and Ribosome Profiling

RPF extraction and sequencing were performed by a commercial company (Chi-Biotech, Wuhan, China) according to a previous study [17]. Spleen tissue from each group was added to lysis buffer, ground at low temperature and then low concentrations of RNase I were added for digestion. The digested samples were pooled and layered on the surface of 15 mL sucrose buffer (30% sucrose in RB buffer). The ribosomes were pelleted by ultracentrifugation at 42,500 rpm for 5 h at 4 °C. RPF extraction was then performed using TRIzol, and ribosomal RNA (rRNA) was depleted using the Ribo-off^®^ rRNA Depletion Kit (Vazyme) following the manufacturer’s instructions. Sequencing libraries of RPFs were constructed following the VAHTS^®^ Small RNA Library Prep Kit for Illumina (Vazyme). The library was resolved by a 6% polyacrylamide gel. The fraction with an insertion size of ~28 nt was excised and purified from the gel. This fraction was sequenced by an Illumina HiSeq-2000 sequencer for 50 cycles. High-quality reads that passed the Illumina quality filters were kept for sequence analysis.

### 4.5. Sequence Analysis

For both mRNA and RPF sequencing data sets, high-quality reads were mapped to the mRNA reference sequence (GRCg6a) through the FANSe2 algorithm [48] with the parameters -E5% --indel -S14. The expression abundance of mRNA and RPFs was normalized by RPKM (reads per kilobase per million reads) [49]. Differentially expressed genes (DEGs) in RNA-seq and Ribo-seq were identified via the edgeR package [50] with |log_2_ fold change| > 1 and false discovery rate (FDR) < 0.01. The quotient of RPFs and mRNA expression abundance is translation efficiency (TE) [51,52]. Differential TE genes (DTEGs) were calculated by a *t* test with |log_2_ fold change| > 1 and *p* value < 0.05. Bioinformatic analysis was performed using Omicsmart, a real-time, interactive online platform for data analysis (http://www.omicsmart.com, accessed on 20 November 2023).

### 4.6. Overexpression of IL4I1 Protein

The recombinant plasmid pEF1α-Myc-IL4I1 was constructed from the IL4I1 gene sequence (Genbank accession number NM_001099351.3) from chicken. DF1 cells were cultured in 6-well plates. When the cell confluency reached 70–80%, the recombinant IL4I1 plasmid was transfected with liposome Lipofectamine^TM^ 3000 (Invitrogen) to overexpress IL4I1 protein. After 24 h of transfection, DF1 cells were infected with the ARV S1133 strain at a multiplicity of infection (MOI) of 1. Then, the sample of cells and medium supernatant were gathered at 24 h postinfection. RNA was extracted from the above cell samples and reverse-transcribed to synthesize cDNA using the GeneJET RNA Purification Kit and Maxima^TM^ H minus cDNA synthesis master mix (Thermo Fisher Scientific). Real-time PCR detected the replication of ARV at the gene level and the expression changes in innate immune signaling pathway-related molecules. The primer sequences of molecules associated with the innate immune signaling pathway and ARV σC gene are shown in Table 1 [53]. In addition, the above medium supernatant was used to infect DF1 cells and the virulence was determined by the Reed–Muench method to detect the replication of ARV.

### 4.7. RNA Interference Assay

According to the sequence of IL4I1 genes, three small interfering RNAs (siRNAs) (Table 2) were designed and synthesized (GenePharma, Shanghai, China). IL4I1 siRNA inhibitory molecules were transfected into DF1 cells using Lipofectamine^®^ RNAiMAX Reagent (Invitrogen) to inhibit IL4I1 protein expression. After 24 h of transfection, cells were infected with the ARV S1133 strain at a MOI of 1, and after another 24 h, cell samples and medium supernatants were collected to detect replication of ARV virus.

### 4.8. Real-Time PCR

Real-time PCR was performed using the PowerUpTM SYBRTM Green Master Mix (Thermo Scientific) via the QuantStudio 5 real-time PCR system (Thermo Lifetech ABI, Boston, MA, USA).

### 4.9. Co-immunoprecipitation (Co-IP) Assays

The interactions between IL4I1 and ARV σA or σC protein were detected by Co-IP. We designed three experimental groups: the test group was cotransfected with pEF1α-Myc-IL4I1 and pEF1α-HA-σA/σC and the control group was cotransfected with pEF1α-Myc-IL4I1 and pEF1α-HA or pEF1α-Myc and pEF1α-HA-σA/σC. Three biological replicates were assigned to each group. After 24 h of transfection, cells were lysed, protein samples were collected and Co-IP analysis with the Pierce Classic IP Kit (Thermo Scientific) was performed. The forward Co-IP test used rabbit-derived HA antibody (Invitrogen) for fixed adsorption samples, and Western blot analysis used murine Myc antibody and murine HA antibody (Invitrogen) as primary antibodies for detection. The reverse Co-IP used rabbit-derived Myc antibody (Invitrogen) for fixed adsorbed samples, and Western blot analysis used murine HA antibody and murine Myc antibody (Invitrogen) as primary antibodies. Goat anti-murine IgG (Beyotime, Shanghai, China) labeled with alkaline phosphatase was the secondary antibody.

### 4.10. Statistical Analysis

At least three valid repeat tests were performed for each treatment, and the results are expressed as the mean ± SD. Graph analysis and statistical comparisons used GraphPad Prism statistical software, version 9.5.0. The unpaired two tailed *t*-test (for two groups) and one-way ANOVA (for multiple groups) were used to identify the significance of difference. The results’ difference were considered statistically significant at *p* < 0.05.

## 5. Conclusions

In this study, we determined that the spleen produces a strong innate immune response at both the transcriptional and translational levels after ARV infection and that the spleen is an important immune response organ in ARV infection. ARV infection reduces the translation efficiency of innate immunity-related genes, and we speculate that the decrease in translation efficiency is the key cause of immunosuppression caused by ARV infection. ARV infection can significantly upregulate the expression of IL4I1, while the upregulation of IL4I1 helps to inhibit the replication of ARV. IL4I1 may inhibit the innate immune response triggered by ARV infection. In addition, the IL4I1 protein may interact with the viral protein σA of ARV. These results provide new insights into ARV–host interactions and will facilitate the development of new vaccines or other therapeutic agents to control ARV based on the IL4I1 gene in chickens.

## Figures and Tables

**Figure 1 viruses-15-02346-f001:**
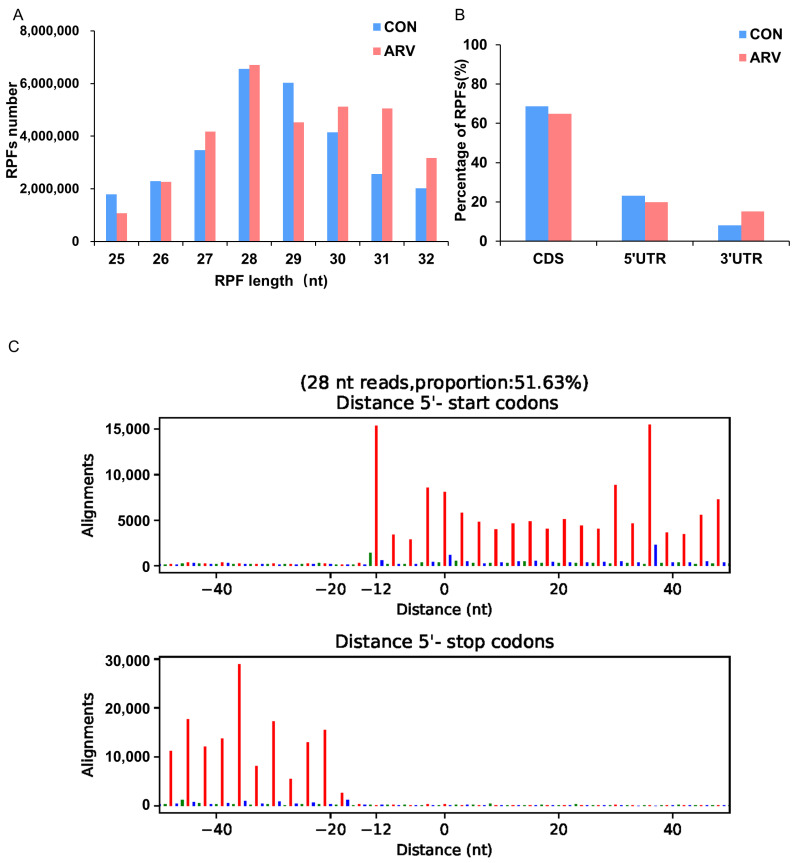
Characteristics of ribosome profiling data in the CON and ARV groups. (**A**) Length distribution of RPFs. (**B**) The percentage of RPFs located in CDS, 5′ UTR and 3′ UTR. (**C**) The total number of RPFs along CDS start and stop codon regions in ARV group. A codon contains three bases, which are represented by red, blue and green.

**Figure 2 viruses-15-02346-f002:**
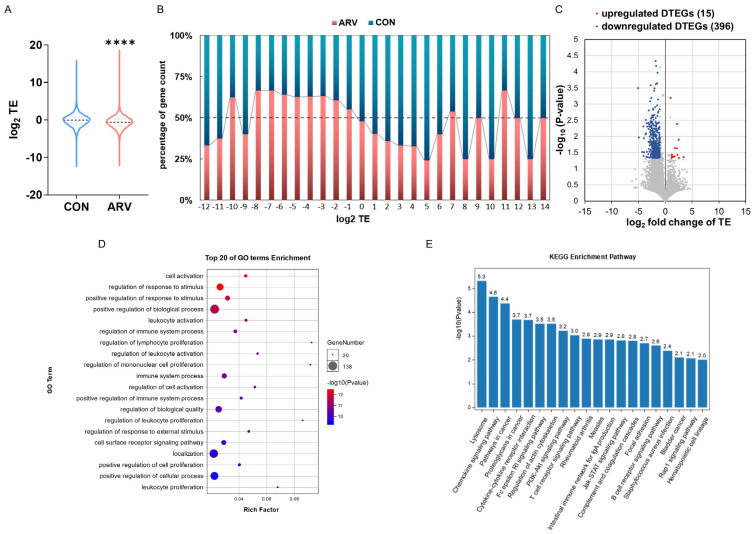
Translation efficiency analysis. (**A**) Violin plot of TE in the CON and ARV groups. Asterisk indicates significant difference (**** *p* < 0.0001). (**B**) Relative TE ratio. Genes were classified based on their rounded log_2_ TE values. (**C**) Volcano map of DTEGs. The gray dot represents the non-differentially expressed translation efficiency genes. (**D**) Gene ontology (GO) analysis of DTEGs. (**E**) Kyoto Encyclopedia of Genes and Genomes (KEGG) analysis of DTEGs.

**Figure 3 viruses-15-02346-f003:**
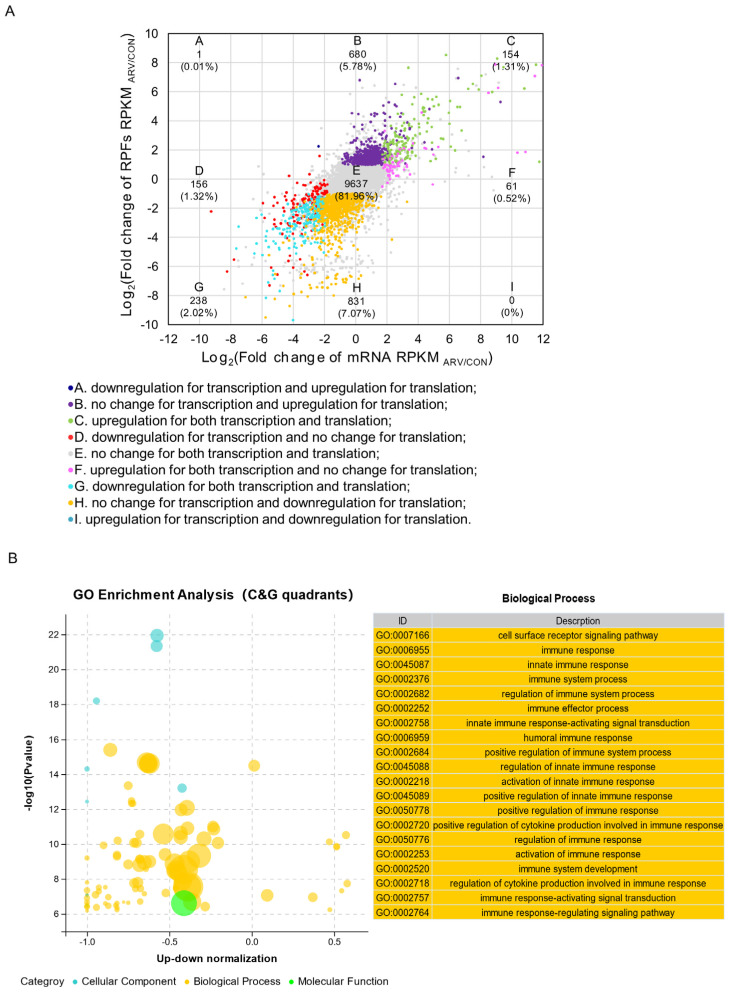
Avian reovirus altered gene expression at both the transcriptional and translational levels. (**A**) Scatter plot of the fold change in the ARV/CON group at the transcriptional and translational levels. Nine squares in different colors indicate nine response groups (|log_2_ fold change| ≥ 1 and FDR < 0.01). (**B**) GO enrichment analysis of genes in quadrants C and G.

**Figure 4 viruses-15-02346-f004:**
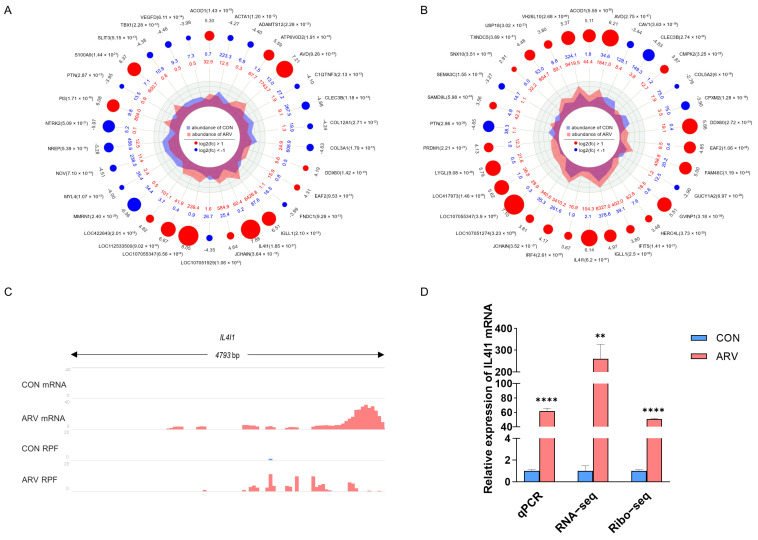
Functional gene screening. The circle maps of the top 30 DEGs in the transcriptome (**A**) and translatome (**B**). (**C**) The expression abundance of IL4I1 at the transcriptional and translational levels. (**D**) The relative level of IL4I1 mRNA expression. Asterisks indicate significant differences (** *p* < 0.01, **** *p* < 0.0001).

**Figure 5 viruses-15-02346-f005:**
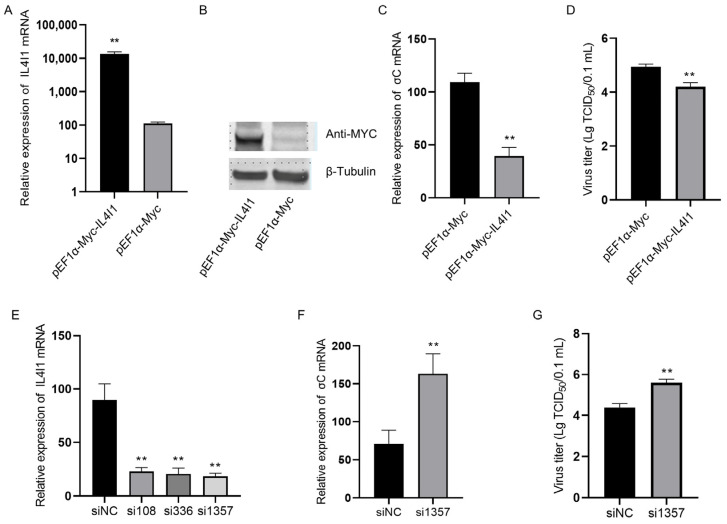
IL4I1 inhibits the replication of ARV in DF1 cells. DF-1 cells were transfected with pEF1α-Myc-IL4I1 or pEF1α-Myc plasmids, and both real-time PCR (**A**) and Western blot (**B**) confirmed high levels of IL4I1 expression in DF-1 cells. DF-1 cells were transfected with pEF1α-Myc-IL4I1 or pEF1α-Myc plasmids and infected with the ARV S1133 strain at an MOI of 1. Viral replication was detected by real-time PCR (**C**) and viral titer (**D**) 24 h postinfection. Comparison of the inhibition efficiency of three siRNAs of IL4I1 by real-time PCR (**E**). DF1 cells were transfected with si1357 or siNC and infected with the ARV S1133 strain of virus at an MOI of 1. Viral replication was detected by real-time PCR (**F**) and viral titer (**G**) 24 h postinfection. Data are represented as the mean ± SD of three independent experiments. Asterisks indicate significant differences (** *p* < 0.01).

**Figure 6 viruses-15-02346-f006:**
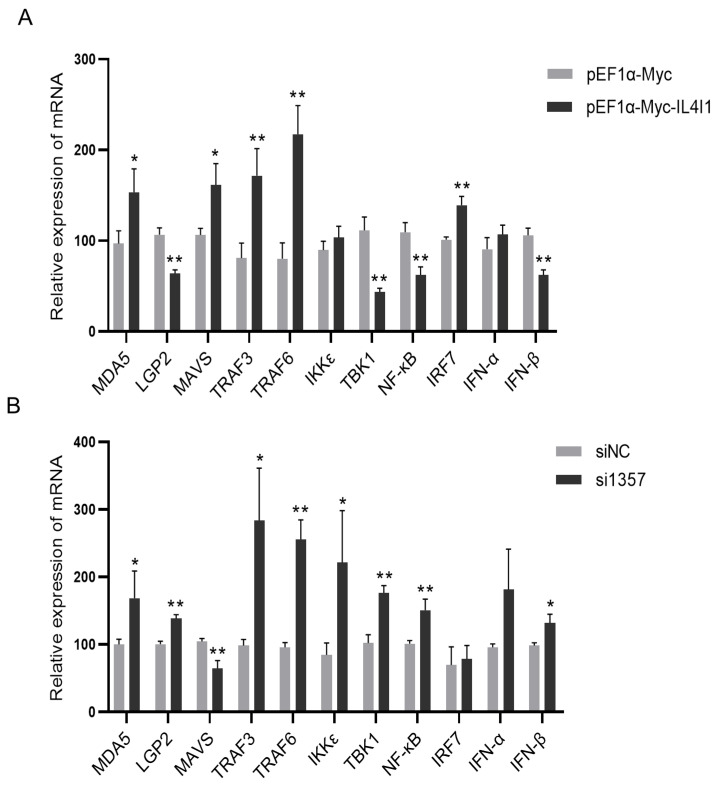
Effect of IL4I1 on the innate immune response during ARV infection. DF-1 cells were transfected with pEF1α-Myc-IL4I1 or pEF1α-Myc plasmids and then infected with the ARV S1133 strain at an MOI of 1. Expression changes in genes associated with the innate immune signaling pathway were detected by real-time PCR (**A**) 24 h postinfection. DF1 cells were transfected with si1357 or siNC and infected with the ARV S1133 strain of virus at an MOI of 1. Expression changes in genes associated with the innate immune signaling pathway were detected by real-time PCR (**B**) 24 h postinfection. Data are represented as the mean ± SD of three independent experiments. Asterisks indicate significant differences (* *p* < 0.05, ** *p* < 0.01).

**Figure 7 viruses-15-02346-f007:**
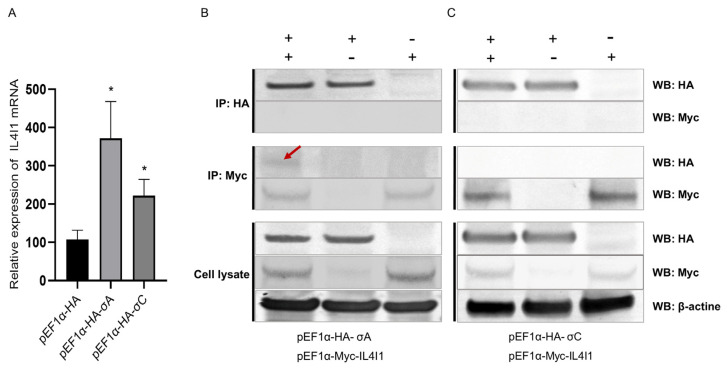
IL4I1 in DF1 cells interacts with ARV σA and σC proteins. PEF1α-HA-σA or pEF1α-HA-σC were transfected into DF1 cells to overexpress ARV σA or σC proteins, respectively. Determination of IL4I1 expression by real-time PCR (**A**). We cotransfected cells with pEF1α-Myc-IL4I1 and pEF1α-HA-σA/σC in the experimental group and pEF1α-Myc-IL4I1 and pEF1α-HA or pEF1α-Myc and pEF1α-HA-σA/σC in the control group. After 24 h of transfection, cells were lysed to collect protein samples, and the interaction of IL4I1 with ARV σA and σC proteins in DF-1 cells was verified by Co-IP. Verification of IL4I1 interaction with ARV σA by Co-IP (**B**). Verification of IL4I1 interaction with ARV σC by Co-IP (**C**). Asterisks indicate significant differences (* *p* < 0.05).

**Table 1 viruses-15-02346-t001:** Primers used in this study.

Gene	Genbank Accession Number	Primer Sequences (5′-3′)
ARV σC	L39002.1	F: CCACGGGAAATCTCACGGTCACT, R: TACGCACGGTCAAGGAACGAATGT
IL4I1	NM_001099351.3	F: CACGCCGTATCAGTTCACC, R: CCTCACCGCAGCCTTCAT
IFN-α	AB021154.1	F: ATGCCACCTTCTCTCACGAC, R: AGGCGCTGTAATCGTTGTCT
IFN-β	X92479.1	F: ACCAGGATGCCAACTTCT, R: TCACTGGGTGTTGAGACG
MDA5	NM_001193638	F: CAGCCAGTTGCCCTCGCCTCA, R: AACAGCTCCCTTGCACCGTCT
LGP2	MF563595.1	F: CCAGAATGAGCAGCAGGAC, R: AATGTTGCACTCAGGGATGT
MAVS	MF289560.1	F: CCTGACTCAAACAAGGGAAG, R: AATCAGAGCGATGCCAACAG
TRAF3	XM_040672281.1	F: GGACGCACTTGTCGCTGTTT, R: CGGACCCTGATCCATTAGCAT
TRAF6	XM_040673314.1	F: GATGGAGACGCAAAACACTCAC, R: GCATCACAACAGGTCTCTCTTC
IKKε	XM_428036.4	F: TGGATGGGATGGTGTCTGAAC, R: TGCGGAACTGCTTGTAGATG
TBK1	MF159109.1	F: AAGAAGGCACACATCCGAGA, R: GGTAGCGTGCAAATACAGC
IRF7	NM_205372.1	F: CAGTGCTTCTCCAGCACAAA, R: TGCATGTGGTATTGCTCGAT
NF-κB	NM_205129.1	F: CATTGCCAGCATGGCTACTAT, R: TTCCAGTTCCCGTTTCTTCAC
GAPDH	NM_204305.1	F: GCACTGTCAAGGCTGAGAACG, R: GATGATAACACGCTTAGCACCAC

**Table 2 viruses-15-02346-t002:** siRNA sequence targeting the IL4I1 gene.

siRNA	Sense	Antisense
si108	GCUGCUGAGUAUUGUGAAATT	UUUCACAAUACUCAGCAGCTT
si336	GCUGGUGCGUGAGUUUAUATT	UAUAAACUCACGCACCAGCTT
si1357	CCGUAUCAGUUCACCGAUUTT	AAUCGGUGAACUGAUACGGTT
siNC	UUCUCCGAACGUGUCACGUTT	ACGUGACACGUUCGGAGAATT

## Data Availability

The data sets presented in this study can be found in online repositories. The names of the repository/repositories and accession number(s) can be found below: https://www.ncbi.nlm.nih.gov/geo/query/acc.cgi?acc=GSE241418, accessed on 20 November 2023.

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
