# Peer review of "Transcriptomic and Translatomic Analyses Reveal Insights into the Signaling Pathways of the Innate Immune Response in the Spleens of SPF Chickens Infected with Avian Reovirus"

_viruses, 2023, doi:10.3390/v15122346_

Round 1
Reviewer 1 Report
Comments and Suggestions for Authors
Wang et al., examined the gene and protein expression patters in spleen of chicken infected with avian reovirus (ARV) strain S1133. Using mRNA sequencing (RNA-seq) and ribosomal foot print sequencing (Ribo-seq) technology, the authors revealed the host protein expressions affected by ARV at transcriptional and translational level. Among the numerous RNA-seq and Ribo-seq data, the authors found the Interleukin 4I1 (IL4I1) was enhanced by ARV infection. Subsequently, using DF1 cell line, it was shown that overexpression of IL4I1 inhibited replication of ARV. Further, the authors showed the evidence that interaction of IL4I1 and sigmaA protein of ARV might be a molecular mechanism of viral replication inhibition by IL4I1.
The result clearly indicated that ARV infection led to elevation of IL4I1 and overexpression of IL4I1 inhibited the AVR replication. Although the biological significance of IL4I1 in ARV infection needs to be examine carefully in future study and the molecular interactions between IL4I1, viral infection/replication, interferon signals and viral proteins were poorly investigated, the study has significant importance to understand host gene reactions to ARV infection and involvement of IL4I1.
Minor comments
1. What organs/cells does ARV infect in chicken? Does ARV infect cells in spleen? It was unclear whether the alteration of protein expression in splenic cells were induced by direct ARV infection or secondary reaction to cytokines released by other virus-infected cells. The link between the in vivo data and the experiment using DF1 cells should be discussed.
2. In figure 6. Please clarify whether IL4I1 suppresses TBK1/LGP2/IFN-beta directly or inhibition of ARV replication by IL4I1 leads to the reduction of TBK1/LGP2/IFN-beta.
3. The interaction of IL4I1 and sigmaA appears to be less clear. In Figure 7B, the band indicated by red arrow looked quite thin. In figure 8, both IL4I1 and sigmaA localized broadly in the cytoplasm and was hard to conclude if they were co-localized. In the manuscript, it was not clear why the authors tested sigmaA and sigmaC. It is recommended to examine other viral proteins.
4. Does ARV infection induce IL4I1 in DF1 cells?
5. Line 115. Show full spelling of ‘CON’ on initial appearance.
6. Figure 5D and G. Data should be shown in bars.
7. Figure 7. Align figure captions properly (i.e. WB:HA, WB:Myc etc. at right)
Author Response
Summary
Wang et al., examined the gene and protein expression patters in spleen of chicken infected with avian reovirus (ARV) strain S1133. Using mRNA sequencing (RNA-seq) and ribosomal foot print sequencing (Ribo-seq) technology, the authors revealed the host protein expressions affected by ARV at transcriptional and translational level. Among the numerous RNA-seq and Ribo-seq data, the authors found the Interleukin 4I1 (IL4I1) was enhanced by ARV infection. Subsequently, using DF1 cell line, it was shown that overexpression of IL4I1 inhibited replication of ARV. Further, the authors showed the evidence that interaction of IL4I1 and sigmaA protein of ARV might be a molecular mechanism of viral replication inhibition by IL4I1.
The result clearly indicated that ARV infection led to elevation of IL4I1 and overexpression of IL4I1 inhibited the AVR replication. Although the biological significance of IL4I1 in ARV infection needs to be examine carefully in future study and the molecular interactions between IL4I1, viral infection/replication, interferon signals and viral proteins were poorly investigated, the study has significant importance to understand host gene reactions to ARV infection and involvement of IL4I1.
Minor comments
Response: We feel great thanks for your professional review work on our article. As you are concerned, there are several problems that need to be addressed. According to your nice suggestion, we have made corrections to our previous draft, the detailed corrections are listed below.
Minor comments
- What organs/cells does ARV infect in chicken? Does ARV infect cells in spleen? It was unclear whether the alteration of protein expression in splenic cells were induced by direct ARV infection or secondary reaction to cytokines released by other virus-infected cells. The link between the in vivo data and the experiment using DF1 cells should be discussed.
Response: We thank for the point raised from the reviewer.
After ARV infection, ARV replication can be detected in various tissues and organs of chickens, such as joints, spleen, bursa of Fabricius, thymus, and peripheral blood lymphocytes [1-3].
Previous studies have found that ARV infection can cause harm to the spleen, which is leads to immunosuppression [4-5]. The results of our previous study showed that the viral load in the spleen after ARV infection was noticeably above those in the thymus and bursa of Fabricius, suggesting that the spleen is the main immune organ attacked by ARV. In addition, the mRNA expression of various interferon-stimulated genes in the spleen after ARV infection was rapidly upregulated in the early stage of infection, indicating that ARV infection can induce a strong innate immune response in the spleen [3].
In this experiment, we used SPF chickens, which do not contain other pathogenic microorganisms and were strictly raised with SPF isolators. Therefore, the pathological changes in the spleen after infection and the changes in protein expression were caused by ARV infection.
ARV can replicate in DF1 cells and cause changes in the expression of innate immune-related factors. At present, DF1 cells are used in vitro to study the pathogenic mechanism of ARV [6-8]. DF1 cells were also used in our experiments to study the role of IL4I1 in ARV replication.
References:
[1]. Xie, L.; Xie, Z.; Wang, S.; Huang, J.; Deng, X.; Xie, Z.; Luo, S.; Zeng, T.; Zhang, Y.; Zhang, M. Altered gene expression pro-files of the MDA5 signaling pathway in peripheral blood lymphocytes of chickens infected with avian reovirus. Arch. Virol. 2019, 164, 2451-2458, doi:10.1007/s00705-019-04340-8.
[2] Wang, S.; Xie, L.; Xie, Z.; Wan, L.; Huang, J.; Deng, X.; Xie, Z.Q.; Luo, S.; Zeng, T.; Zhang, Y. et al. Dynamic Changes in the Expression of Interferon-Stimulated Genes in Joints of SPF Chickens Infected With Avian Reovirus. Front Vet Sci 2021, 8, 618124, doi:10.3389/fvets.2021.618124.
[3]. Wang, S.; Wan, L.; Ren, H.; Xie, Z.; Xie, L.; Huang, J.; Deng, X.; Xie, Z.; Luo, S.; Li, M. et al. Screening of interfer-on-stimulated genes against avian reovirus infection and mechanistic exploration of the antiviral activity of IFIT5. Front. Microbiol. 2022, 13, 998505, doi:10.3389/fmicb.2022.998505.
[4] Rosenberger, J.K.; Sterner, F.J.; Botts, S.; Lee, K.P.; Margolin, A. In vitro and in vivo characterization of avian reoviruses. I. Pathogenicity and antigenic relatedness of several avian reovirus isolates. Avian Dis. 1989, 33, 535-544.
[5] Roessler, D.E.; Rosenberger, J.K. In vitro and in vivo characterization of avian reoviruses. III. Host factors affecting virulence and persistence. Avian Dis. 1989, 33, 555-565.
[6] Xie, L.; Xie, Z.; Huang, L.; Fan, Q.; Luo, S.; Huang, J.; Deng, X.; Xie, Z.; Zeng, T.; Zhang, Y. et al. Avian reovirus sigmaA and sigmaNS proteins activate the phosphatidylinositol 3-kinase-dependent Akt signalling pathway. Arch. Virol. 2016, 161, 2243-2248, doi:10.1007/s00705-016-2908-6.
[7] Xie, L.; Wang, S.; Xie, Z.; Wang, X.; Wan, L.; Deng, X.; Xie, Z.; Luo, S.; Zeng, T.; Zhang, M. et al. Gallus NME/NM23 nucleo-side diphosphate kinase 2 interacts with viral sigmaA and affects the replication of avian reovirus. Vet. Microbiol. 2021, 252, 108926, doi:10.1016/j.vetmic.2020.108926.
[8] Zhang, Z.; Lin, W.; Li, X.; Cao, H.; Wang, Y.; Zheng, S.J. Critical role of eukaryotic elongation factor 1 alpha 1 (EEF1A1) in avian reovirus sigma-C-induced apoptosis and inhibition of viral growth. Arch. Virol. 2015, 160, 1449-1461, doi:10.1007/s00705-015-2403-5.
- In figure 6. Please clarify whether IL4I1 suppresses TBK1/LGP2/IFN-beta directly or inhibition of ARV replication by IL4I1 leads to the reduction of TBK1/LGP2/IFN-beta.
Response: Thank you for this valuable question, your question is very good.
The current studies have shown that IL4I1 is an important immunosuppressive molecule that plays a key role in the immune evasion of tumors [1]. Previous studies have also found that IL4I1 can inhibit the production of IFN-γ and inflammatory cytokines, limit local Th1 inflammation, and inhibit the inflammatory response [2]. In our research, investigation on the role of IL4I1 in ARV infection showed that overexpression of IL4I1 gene inhibited the replication of ARV in DF1, whereas inhibition of endogenously expressed IL4I1 gene by siRNA facilitated ARV replication. Real-time quantitative PCR was used to evaluate the effects of IL4I1 on the expression of related factors in the natural immune signaling pathway during ARV infection. We found that the mRNA expression levels of IFN-β, LGP2, TBK1 and NF-κB genes were significantly down-regulated after overexpression of IL4I1 gene, but significantly up-regulated after inhibition of IL4I1 gene expression. Therefore, IL4I1 is likely to be a negative feedback regulator of natural immune signaling pathway. IL4I1 may reduce IFN-β production by inhibiting the expression of LGP2, TBK1 and NF-κB genes.
As an immunomodulator, whether IL4I1 directly inhibits the expression of these factors and exerts immunomodulatory effects during ARV infection, or inhibits the expression of these factors by inhibiting the expression of ARV, these will be the contents of our future research. Recently, we have received the support from a new project to investigate the mechanism of action of IL4I1 in ARV infection.
References:
[1]. Sadik, A.; Somarribas, P.L.; Ozturk, S.; Mohapatra, S.R.; Panitz, V.; Secker, P.F.; Pfander, P.; Loth, S.; Salem, H.; Prentzell, M.T. et al. IL4I1 Is a Metabolic Immune Checkpoint that Activates the AHR and Promotes Tumor Progression. Cell 2020, 182, 1252-1270, doi:10.1016/j.cell.2020.07.038.
[2]. Marquet, J.; Lasoudris, F.; Cousin, C.; Puiffe, M.L.; Martin-Garcia, N.; Baud, V.; Chereau, F.; Farcet, J.P.; Molinier-Frenkel, V.; Castellano, F. Dichotomy between factors inducing the immunosuppressive enzyme IL-4-induced gene 1 (IL4I1) in B lymphocytes and mononuclear phagocytes. Eur. J. Immunol. 2010, 40, 2557-2568, doi:10.1002/eji.201040428.
- The interaction of IL4I1 and sigmaA appears to be less clear. In Figure 7B, the band indicated by red arrow looked quite thin. In figure 8, both IL4I1 and sigmaA localized broadly in the cytoplasm and was hard to conclude if they were co-localized. In the manuscript, it was not clear why the authors tested sigmaA and sigmaC. It is recommended to examine other viral proteins.
Response: Thank you for this valuable question. Other studies have found that the σA and σC proteins of ARV play a crucial part in virus interacts with its host. A study found that the ARV σA protein binds irreversibly to viral dsRNA, thereby inhibiting the dsR-NA-dependent protein kinases activation and ultimately interfering the antiviral effects of interferon [1-2]. In addition, the ARV σA protein can also activate the PI3K/Akt signal transduction pathways in cells, increase the expression of phosphorylated Akt (p-Akt) in cells, and thus inhibit the apoptosis of infected cells to facilitate ARV infection and repli-cation [3]. It has also been showed that the ARV σA protein affects the replication of ARV in DF1 cells by interacting with the NME2 protein of the host [4]. The ARV σC protein is related to the adsorption and proliferation of virions [5]. The ARV σC protein is able to induce apoptosis by interacting with the host protein EFF1A1 [6]. Therefore, we chose to analyze the interaction of σA and σC proteins with IL4I1 proteins.
The interaction of IL4I1 and σA appears to be less clear. To ensure the rigor of the experimental data, we performed multiple replicates using antibodies and Co-IP kits of different brands, all with the same results.
As for the result in Figure 8, combined with reviewer 2's opinion, we have decided to remove this section.
References:
[1] Vazquez-Iglesias, L.; Lostale-Seijo, I.; Martinez-Costas, J.; Benavente, J. Avian reovirus sigmaA localizes to the nucleolus and enters the nucleus by a nonclassical energy- and carrier-independent pathway. J. Virol. 2009, 83, 10163-10175, doi:10.1128/JVI.01080-09.
[2] Gonzalez-Lopez, C.; Martinez-Costas, J.; Esteban, M.; Benavente, J. Evidence that avian reovirus sigmaA protein is an in-hibitor of the double-stranded RNA-dependent protein kinase. J. Gen. Virol. 2003, 84, 1629-1639, doi:10.1099/vir.0.19004-0.
[3] Xie, L.; Xie, Z.; Huang, L.; Fan, Q.; Luo, S.; Huang, J.; Deng, X.; Xie, Z.; Zeng, T.; Zhang, Y. et al. Avian reovirus sigmaA and sigmaNS proteins activate the phosphatidylinositol 3-kinase-dependent Akt signalling pathway. Arch. Virol. 2016, 161, 2243-2248, doi:10.1007/s00705-016-2908-6.
[4] Xie, L.; Wang, S.; Xie, Z.; Wang, X.; Wan, L.; Deng, X.; Xie, Z.; Luo, S.; Zeng, T.; Zhang, M. et al. Gallus NME/NM23 nucleo-side diphosphate kinase 2 interacts with viral sigmaA and affects the replication of avian reovirus. Vet. Microbiol. 2021, 252, 108926, doi:10.1016/j.vetmic.2020.108926.
[5] Shmulevitz, M.; Yameen, Z.; Dawe, S.; Shou, J.; O'Hara, D.; Holmes, I.; Duncan, R. Sequential partially overlapping gene arrangement in the tricistronic S1 genome segments of avian reovirus and Nelson Bay reovirus: implications for transla-tion initiation. J. Virol. 2002, 76, 609-618, doi:10.1128/jvi.76.2.609-618.2002.
[6] Zhang, Z.; Lin, W.; Li, X.; Cao, H.; Wang, Y.; Zheng, S.J. Critical role of eukaryotic elongation factor 1 alpha 1 (EEF1A1) in avian reovirus sigma-C-induced apoptosis and inhibition of viral growth. Arch. Virol. 2015, 160, 1449-1461, doi:10.1007/s00705-015-2403-5.
- Does ARV infection induce IL4I1 in DF1 cells?
Response: Thank you for your comment. ARV infection also induces up-regulated expression of IL4I1 gene in DF1.
- Line 115. Show full spelling of ‘CON’ on initial appearance.
Response: Thank you for this valuable question. “CON” is an abbreviation for the “control group”, It has been added to the manuscript.
- Figure 5D and G. Data should be shown in bars.
Response: Thank you for your comment. Figure 5D and G It has been changed to a histogram.
- Figure 7. Align figure captions properly (i.e. WB:HA, WB:Myc etc. at right)
Response: Thanks for your comment, the Figure 7 has been revised.
Reviewer 2 Report
Comments and Suggestions for Authors
Summary
In this manuscript, the authors examined both the transcriptome and translatome of cellular genes following avian reovirus infection. Avian reovirus is clearly a major cause of avian diseases, yet detailed data on the transcriptome and translatome from organs of infected animals have not been studied in detail before. The topic is thus quite relevant for the journal, especially in the context of this issue. The authors do present in detail in the introduction why their study is novel, compared to previous works. In their study, they concentrate their efforts on the spleen of infected animals, that seems like a logical choice.
General comments
•I am not convinced that ribosomal protection is the best approach compared to ribosomal profiling. I suppose that this was chosen since it is from a tissue and ribosomal profiling is probably challenging. The authors could still mention it for the benefit of the reader.
•I easily found published manuscript on the transcriptome of avian reovirus infected cells and of Muscovy duck reovirus infected tissues. Although I agree that the authors present something more complete they should also acknowledge previous work that could be pertinent and compare their data with previous publications.
•If I understand well, the RNAseq and RiboSeq were performed on only two biological replicates. This does not seem much. I think it is quite accepted in the literature that a minimum of three biological replicates is necessary in these experiments and in the statistical analysis that ensues. Could the authors add more precision to explain the rationale of using 2 samples and the rational of their statistical analysis. I am not an expert on this specific aspect, but I do not think that a Student’s t test is appropriate. If it is not already the case, the authors should consult a biostatistician or a bioinformatician.
•On figure 7, the experiment with sigmaA is quite convincing but the lack of interaction with sigmaC is less convincing since the immunoblots are much less intense, raising the possibility that the signal was missed. Also, how many times was the overall experiment repeated?
•As far as I know, the IL4I1 protein is normally secreted whereas the authors have shown it to be cytoplasmic. The authors do not comment on the known, or expected, localization of the protein in the literature. The interaction of sigma1 with IL4I1 is well shown in the co-IP experiment but I am not convinced that the co-localization experiment is that convincing. Obviously, two proteins with a cytoplasmic localization will be seen to co-localize to a large extent. Also, essentially one cell is presented without quantification on a large number of cells. Overall, this was the last convincing date of the paper, this could possibly be removed without affecting my overall judgment on the manuscript.
Minor comments
•Although the manuscript is submitted to Viruses, the form used is from Int. J. Mol. Sci.
•Line 41 : According to the most recent ICTV classification, the family is Spinareoviridae
Author Response
Summary
In this manuscript, the authors examined both the transcriptome and translatome of cellular genes following avian reovirus infection. Avian reovirus is clearly a major cause of avian diseases, yet detailed data on the transcriptome and translatome from organs of infected animals have not been studied in detail before. The topic is thus quite relevant for the journal, especially in the context of this issue. The authors do present in detail in the introduction why their study is novel, compared to previous works. In their study, they concentrate their efforts on the spleen of infected animals, that seems like a logical choice.
Response: Thank you very much for your professional review work on our article. We have carefully considered all your suggestions and made revisions, hoping to resolve your concerns. In the remainder of this letter, we will discuss each of your comments and respond accordingly.
General comments
- I am not convinced that ribosomal protection is the best approach compared to ribosomal profiling. I suppose that this was chosen since it is from a tissue and ribosomal profiling is probably challenging. The authors could still mention it for the benefit of the reader.
Response: We thank for the point raised from the reviewer. Ribosome profiling and ribosomal footprint sequencing are different names for ribo-seq and are actually the same concept [1-3]. We have optimized the wording of this place to avoid misunderstanding. (Line 78).
References:
[1] Chothani SP, Adami E, Widjaja AA, Langley SR, Viswanathan S, Pua CJ, Zhihao NT, Harmston N, D'Agostino G, Whiffin N, Mao W, Ouyang JF, Lim WW, Lim S, Lee CQE, Grubman A, Chen J, Kovalik JP, Tryggvason K, Polo JM, Ho L, Cook SA, Rackham OJL, Schafer S. A high-resolution map of human RNA translation. Mol Cell. 2022 Aug 4;82(15):2885-2899.e8. doi: 10.1016/j.molcel.2022.06.023. Epub 2022 Jul 15. PMID: 35841888.
[2] Manne BK, Campbell RA, Bhatlekar S, Ajanel A, Denorme F, Portier I, Middleton EA, Tolley ND, Kosaka Y, Montenont E, Guo L, Rowley JW, Bray PF, Jacob S, Fukanaga R, Proud C, Weyrich AS, Rondina MT. MAPK-interacting kinase 1 regulates platelet production, activation, and thrombosis. Blood. 2022 Dec 8;140(23):2477-2489. doi: 10.1182/blood.2022015568. PMID: 35930749; PMCID: PMC9918849.
[3] Huang T, Yu L, Pan H, Ma Z, Wu T, Zhang L, Liu K, Qi Q, Miao W, Song Z, Zhang H, Zhou L, Li Y. Integrated Transcriptomic and Translatomic Inquiry of the Role of Betaine on Lipid Metabolic Dysregulation Induced by a High-Fat Diet. Front Nutr. 2021 Oct 11;8:751436. doi: 10.3389/fnut.2021.751436. PMID: 34708066; PMCID: PMC8542779.
- I easily found published manuscript on the transcriptome of avian reovirus infected cells and of Muscovy duck reovirus infected tissues. Although I agree that the authors present something more complete they should also acknowledge previous work that could be pertinent and compare their data with previous publications.
Response: Thank you for this valuable question. We analyzed transcriptome data of avian reovirus infected cells [1] and of Muscovy duck reovirus infected tissues [2]. For the transcriptome data of avian reovirus infected DF1 cells, we found no IL4I1-related information in the differential expression matrix. For the transcriptome data of Muscovy duck reovirus infected liver tissues, the original data format uploaded by the author was not professional, which made it difficult for us to obtain transcriptome information. Considering that different people may have different ways of creating transcriptome libraries and the existence of bulk effects, we will not discuss these two data too much. Most importantly, our experimental results are consistent with the sequencing results, which shows the reliability of this study.
In the discussion section, we discussed that the up regulation of IL4I1 in chicken embryonic fibroblasts, primary mononuclear macrophages and spleen tissues after infection with ALV-J and MDV viruses was consistent with our results. (Line 386-395).
References:
[1] Niu X, Wang Y, Li M, Zhang X, Wu Y. Transcriptome analysis of avian reovirus-mediated changes in gene expression of normal chicken fibroblast DF-1 cells. BMC Genomics. 2017 Nov 25;18(1):911. doi: 10.1186/s12864-017-4310-5. PMID: 29178824; PMCID: PMC5702118.
[2] Wang Q, Liu M, Xu L, Wu Y, Huang Y. Transcriptome analysis reveals the molecular mechanism of hepatic fat metabolism disorder caused by Muscovy duck reovirus infection. Avian Pathol. 2018 Apr;47(2):127-139. doi: 10.1080/03079457.2017.1380294. Epub 2017 Oct 10. PMID: 28911249.
- If I understand well, the RNAseq and RiboSeq were performed on only two biological replicates. This does not seem much. I think it is quite accepted in the literature that a minimum of three biological replicates is necessary in these experiments and in the statistical analysis that ensues. Could the authors add more precision to explain the rationale of using 2 samples and the rational of their statistical analysis. I am not an expert on this specific aspect, but I do not think that a Student’s t test is appropriate. If it is not already the case, the authors should consult a biostatistician or a bioinformatician.
Response: Thank you for this valuable question, you are quite right.
In most literature, the number of biological replicates is at least three. But the study of translatome does not necessarily require three biological repetitions, and there are many high-level articles with only one sample [1-2]. In addition, we have used two biological replicates in our previous work [3-4]. The standard for selecting sequencing samples in this study was to use the two samples whose viral load of spleen was closest to the mean in the group as 2 biological replicates. (Line 477-479). More importantly, the results of this study were verified by experiments. The above content is the rationale of using 2 samples for this study.
You are right about statistical analysis. Due to an oversight in our writing, the error was made, for which we apologize. We redescribed the content of the statistical analysis and checked all the results. (Line 564-565).
References:
[1] Wang T, Cui Y, Jin J, Guo J, Wang G, Yin X, He QY, Zhang G. Translating mRNAs strongly correlate to proteins in a multivariate manner and their translation ratios are phenotype specific. Nucleic Acids Res. 2013 May;41(9):4743-54. doi: 10.1093/nar/gkt178. Epub 2013 Mar 21. PMID: 23519614; PMCID: PMC3643591.
[2] Lian X, Guo J, Gu W, Cui Y, Zhong J, Jin J, He QY, Wang T, Zhang G. Genome-Wide and Experimental Resolution of Relative Translation Elongation Speed at Individual Gene Level in Human Cells. PLoS Genet. 2016 Feb 29;12(2):e1005901. doi: 10.1371/journal.pgen.1005901. PMID: 26926465; PMCID: PMC4771717.
[3] Huang T, Yu L, Pan H, Ma Z, Wu T, Zhang L, Liu K, Qi Q, Miao W, Song Z, Zhang H, Zhou L, Li Y. Integrated Transcriptomic and Translatomic Inquiry of the Role of Betaine on Lipid Metabolic Dysregulation Induced by a High-Fat Diet. Front Nutr. 2021 Oct 11;8:751436. doi: 10.3389/fnut.2021.751436. PMID: 34708066; PMCID: PMC8542779.
[4] Huang T, Yu J, Luo Z, Yu L, Liu S, Wang P, Jia M, Wu T, Miao W, Zhou L, Song Z, Zhang H, Li Y. Translatome analysis reveals the regulatory role of betaine in high fat diet (HFD)-induced hepatic steatosis. Biochem Biophys Res Commun. 2021 Oct 20;575:20-27. doi: 10.1016/j.bbrc.2021.08.058. Epub 2021 Aug 25. PMID: 34454176.
- On figure 7, the experiment with sigmaA is quite convincing but the lack of interaction with sigmaC is less convincing since the immunoblots are much less intense, raising the possibility that the signal was missed. Also, how many times was the overall experiment repeated?
Response: Thank you for your comment. To ensure the rigor of the experimental data, we performed three replicates using an-tibodies and Co-IP kits of different brands, all with the same results. The interaction between IL4I1 and ARV σC proteins was not found by coimmunoprecipi-tation experiments.
- As far as I know, the IL4I1 protein is normally secreted whereas the authors have shown it to be cytoplasmic. The authors do not comment on the known, or expected, localization of the protein in the literature. The interaction of sigma1 with IL4I1 is well shown in the co-IP experiment but I am not convinced that the co-localization experiment is that convincing. Obviously, two proteins with a cytoplasmic localization will be seen to co-localize to a large extent. Also, essentially one cell is presented without quantification on a large number of cells. Overall, this was the last convincing date of the paper, this could possibly be removed without affecting my overall judgment on the manuscript.
Response: Thank you for this valuable question, you are quite right. We have decided to remove this section.
Minor comments
- Although the manuscript is submitted to Viruses, the form used is from Int. J. Mol. Sci.
Response: Thank you for your comment. The manuscript is formatted according to the format requirements of Viruses.
- Line 41: According to the most recent ICTV classification, the family is Spinareoviridae.
Response: Thanks for your comment, the article has been revised (Line 41).